# 5-(*N*-Trifluoromethylcarboxy)aminouracil as a Potential DNA Radiosensitizer and Its Radiochemical Conversion into *N*-Uracil-5-yloxamic Acid

**DOI:** 10.3390/ijms21176352

**Published:** 2020-09-01

**Authors:** Paulina Spisz, Witold Kozak, Lidia Chomicz-Mańka, Samanta Makurat, Karina Falkiewicz, Artur Sikorski, Anna Czaja, Janusz Rak, Magdalena Zdrowowicz

**Affiliations:** Department of Physical Chemistry, Faculty of Chemistry, University of Gdańsk, Wita Stwosza 63, 80-308 Gdańsk, Poland; paulina.rewers@phdstud.ug.edu.pl (P.S.); davelombardo@wp.pl (W.K.); lidia.chomicz-manka@ug.edu.pl (L.C.-M.); samanta.makurat@ug.edu.pl (S.M.); karina.falki@gmail.com (K.F.); artur.sikorski@ug.edu.pl (A.S.); czaja.ania@yahoo.com (A.C.); janusz.rak@ug.edu.pl (J.R.)

**Keywords:** radiosensitizer, uracil derivatives, modified nucleobases, electron-induced degradation, DFT calculations

## Abstract

Hypoxia—a hallmark of solid tumors—dramatically impairs radiotherapy, one of the most common anticancer modalities. The adverse effect of the low-oxygen state can be eliminated by the concomitant use of a hypoxic cell radiosensitizer. In the present paper, we show that 5-(*N*-trifluoromethylcarboxy) aminouracil (CF_3_CONHU) can be considered as an effective radiosensitizer of DNA damage, working under hypoxia. The title compound was synthesized in the reaction of 5-aminouracil and trifluoroacetic anhydride in trifluoroacetic acid. Then, an aqueous and deoxygenated solution of the HPLC purified compound containing *tert*-butanol as a hydroxyl radical scavenger was irradiated with X-rays. Radiodegradation in a 26.67 ± 0.31% yield resulted in only one major product—*N*-uracil-5-yloxamic acid. The mechanism that is possibly responsible for the formation of the observed radioproduct has been elucidated with the use of DFT calculations. The cytotoxic test against the PC3 prostate cancer cell line and HDFa human dermal fibroblasts confirmed the low cytotoxicity of CF_3_CONHU. Finally, a clonogenic assay and flow cytometric analysis of histone H2A.X phosphorylation proved the radiosensitization in vitro.

## 1. Introduction

Tumor hypoxia, defined as a low level of oxygenation, causes resistance to radiotherapy, which is one of the treatments of choice in many types of cancer [1]. In a hypoxic environment, cancer cells are two to three times more resistant to ionizing radiation (IR) than normoxic ones [2]. The oxygen effect is explained by the oxygen fixation hypothesis, which postulates that radical damage produced in DNA by the direct or indirect action of radiation can be repaired under hypoxia or “fixed” by molecular oxygen even at a very small concentration [3,4]. One of the ways to overcome this important impediment to effective cancer therapy is introducing radiosensitizers. Radiosensitizers are agents that sensitize tumor cells to IR. A “true” radiosensitizer meets the stricter criteria of being (i) relatively nontoxic and (ii) selective for tumors. These species may be grouped into three categories based on their structures [5]. First, nanomaterials such as nanometallic materials, and especially gold-based nanoparticles, are promising potential radiosensitizers because of their ability to absorb, scatter and emit a radiation energy, satisfying chemical stability, high biocompatibility and low toxicity [5]. Second, macromolecules such as miRNAs, proteins, peptides and oligonucleotides are also capable of regulating radiosensitivity and can be combined with biological therapy as well as drug delivery [5]. Finally, a number of small molecules have been explored to develop radiosensitizers. The latter group includes oxygen mimetics, repair process inhibitors, apoptosis activators, suppressors of radioprotective substances or substrates for the biosynthesis of modified DNA—modified nucleosides [5].

Another classification of radiosensitizers takes into account only two groups of sensitizers that can be really used in clinical practice, because of their selectivity: (i) hypoxic-cell radiosensitizers and (ii) halogenated pyrimidines. The first group of radiosensitizers enhances the radiation effects only in hypoxic cells (characteristic of tumor tissue) but not in normoxic ones. For this reason, the selectivity toward cancer is due to the difference in oxygenation levels between cancer and normal cells. The selectivity of halogenated pyrimidines results from rapid growth and uncontrolled division of cancer cells, which means a higher degree of incorporation of pyrimidine analogs into the DNA of cancer cells, compared to healthy ones [6].

Uracil derivatives possess exceptional properties, making them especially good agents for the radiation-induced cell killing [7]. Some of them are able to substitute native nucleobases in DNA without affecting its structure and biological functions, and consequently they are incorporated into the DNA of tumor cells during their non-controlled proliferation. This results in a much larger extent of DNA labeling in cancer cells. Additionally, if nucleosides are appropriately modified, their radiosensitizing properties may occur only under hypoxia.

Such a modification should rely on the introduction of suitable substituents in nucleobases, which increases nucleosides’ sensitivity to degradation induced by solvated electrons [7]. These electrons are one of the major products of water radiolysis under hypoxic conditions [8] (in normoxic conditions solvated electrons can be scavenged by oxygen) and remain non-reactive to native DNA under biological conditions [9,10]. Thus, a modified, radiosensitizing nucleobase has usually an electrophilic substituent and undergoes an efficient dissociative electron attachment (DEA) that leaves behind a nucleoside radical, which, in secondary reactions, is able to produce DNA damage leading to cancer cell death [7]. One of the best known radiosensitizers of this type are the halo derivatives of uracil. In the past, it was shown that cells undergoing DNA synthesis are unable to distinguish between thymidine and 5-halogenated 2’-deoxyuridines, such as 5-bromo-2’-deoxyuridine (BrdU). Therefore, such analogues are incorporated into newly synthesized DNA [11,12]. BrdU is a highly electron-affinic molecule, prone to dissociative electron attachment leading to a reactive uracil-5-yl radical via the unstable anion [13,14,15]. The promising characteristics of the electron-induced degradation of halo uracils have been demonstrated many times [16,17,18,19,20]. Recently, several new C5-pyrimidine analogues have been proposed in the context of efficient electron-induced degradation [21,22]. Among others, one could name: 5-thiocyanato-2’-deoxyuridine [23], 5-selenocyanato-2’-deoxyuridine [24], 5-trifluoromethanesulfonyl-2’-deoxyuridine [25] and 5-iodo-4-thio-2’-deoxyuridine [26].

In the present paper, we demonstrate that 5-(*N*-trifluoromethylcarboxy) aminouracil (CF_3_CONHU) can be considered as an effective radiosensitizer. Indeed, the theoretical profile of dissociative electron attachment to CF_3_CONHU, concerning the release of the fluoride anion, suggests that electron-induced damage to the molecule is kinetically and thermodynamically accessible. A steady state radiolysis of the deoxygenated solution of CF_3_CONHU (obtained in the reaction of 5-aminouracil and trifluoroacetic anhydride in trifluoroacetic acid) containing an ^•^OH radical scavenger has been performed. This experiment, coupled with high performance liquid chromatography and mass spectrometry, allowed us to identify the product of the electron-induced degradation of CF_3_CONHU and determine the efficiency of the process. The mechanism responsible for the formation of the observed radioproduct has been proposed with use of quantum chemical calculations. In view of the radiodamage observed in the radiolytic studies, we also performed a cytotoxicity test and clonogenic assay and found out that the treatment of PC3 prostate cancer cells with the studied compound brings about no significant decrease in cell viability, although it results in a reduction of cell survival after exposure to IR. The results of cytometric studies on histone H2A.X phosphorylation suggest that the radiosensitivity of CF_3_CONHU-treated PC3 cells is related to (at least in part) the formation of double-strand breaks (DSBs).

## 2. Results and Discussion

### 2.1. Design and Synthesis of 5-(N-Trifluoromethylcarboxy)aminouracil

The uracil molecule can be transformed into a potential radiosensitizer by introduction of an appropriate substituent which increases the electron affinity of the system [7]. 5-Substituted uracils are by far the most common derivatives of uracil, due to the susceptibility of this position to chemical transformation. Besides halogenation reactions that lead to the C5–X bond (X = F, Cl, Br or I), other substituents can be relatively easily introduced into this nucleobase [27]. For instance, the amination reaction of 5-bromouracil may serve to obtain 5-aminouracil (NH_2_U). However, as indicated by the computational adiabatic electron affinity (AEA), the vulnerability of 5-aminouracil to accept the excess electron is lower than that of uracil itself (AEA = 2.04 for NH_2_U and 2.09 eV for U calculated at the M06-2X/6-31++G (d,p) level). Yet the amino group can be relatively easily converted into a more electrophilic one. Namely, introducing a highly electronegative trifluoroacetyl group onto the -NH_2_ substituent should improve the electron affinity of NH_2_U. Indeed, the AEA of 5-(*N*-trifluoromethylcarboxy) aminouracil calculated at the M06-2X/6-31++G (d,p) level amounts to 2.41 eV, which is only 0.02 eV lower than that of 5-bromouracil (AEA = 2.43 eV at the same level of theory), a well-known radiosensitizer.

It is worth emphasizing that an increased electron affinity itself is not a sufficient condition for a molecule to be an efficient radiosensitizer. Besides the ability to accept an excess electron, which is necessary to trap a solvated electron, a potential radiosensitizer has to be prone to electron attachment-induced dissociation. In Figure 1, the energetic profile (in the free energy scale) for the possible DEA process in CF_3_CONHU, obtained at the DFT level, is depicted. We considered the cleavage of particular bonds—C5–N, N–C, C–CF_3_ and C–F—in the substituent.

As indicated by the values gathered in Figure 1, only the release of the fluoride anion, induced by electron attachment, is possible. The thermodynamic barriers for the dissociation of the remaining bonds—C5–N, C–N and C–CF_3_—span the range between 24.5 and 34.6 kcal mol^−1^, which is far too high to be competitive with the release of F^−^. Undoubtedly, the latter process is associated with a small activation barrier and a slightly positive thermodynamic stimulus of 5.6 kcal mol^−1^. It is worth underlining that at the B3LYP level this driving force amounts to −4.0 kcal mol^−1^, which on the one hand reflects the semi-empirical nature of the DFT approach and on the other hand suggests that the release of fluoride anions from the primary radical anion can be spontaneous.

In view of the promising results described above, we decided to obtain CF_3_CONHU. The synthesis of the above-mentioned compound relies on the acylation reaction, according to Figure 2. The title compound has been obtained via the reaction of 5-aminouracil with trifluoroacetic anhydride (TFAA) in trifluoroacetic acid (TFA) [28].

### 2.2. Crystallography

Single-crystal X-ray diffraction measurements show that CF_3_CONHU crystallized in the monoclinic *C*2/c space group with one molecule in the asymmetric unit (Figure 3 and Appendix A). In the crystal of a title compound, the molecules are linked through N1–H1···O7 and N3–H3···O8 hydrogen bonds to create sheets of asymmetric ribbons (Figure 4 and Appendix A), previously observed in a crystal of form II of 5-fluorouracil [29]. The neighboring, anti-parallel ribbons are connected via N9–H9···O8, C6–H6···O11 hydrogen bonds and F15···F15 contact (d (F···F) = 2.81 Å) to form a 3D framework.

### 2.3. Radiolysis

Two of the most important properties which a DEA type radiosensitizer should possess are (i) high electron affinity of the nucleobase and (ii) the ease of dissociation of the radical anion, formed as a result of electron attachment, leading to reactive radicals that in secondary steps may produce strand breaks [7]. Therefore, the steady state radiolysis has been performed for an aqueous solution of CF_3_CONHU. Thus, the aqueous solution of 5-(*N*-trifluoromethylcarboxy) aminouracil has been irradiated at pH = 7 with a dose of 140 Gy in the presence of phosphate buffer and *tert*-butanol (as ^•^OH radical scavenger). Prior to IR exposure, samples were deoxygenated by Ar saturation in order to avoid the scavenging of electrons by oxygen molecules. In the end, the HPLC and LC-MS analyses of the irradiated and non-irradiated samples were performed.

Figure 5 shows that only one main radioproduct has been produced. Based on the MS and MS/MS spectra, we found that the product of the electron-induced degradation of CF_3_CONHU is *N*-uracil-5-yloxamic acid (for MS and MS/MS spectra with ion identities, see the Appendix A). This product has been generated due to the DEA process, which, in the first step, leads to a fluoride anion detachment and the generation of a ^•^CF_2_CONHU radical (for the detailed mechanism, see Figure 6). The yield of CF_3_CONHU degradation, 26.67 ± 0.31%, has been determined by comparing the areas of its HPLC signals before and after IR exposure. It is worth noting that the decay of BrdU—a well-known radiosensitizer—is equal to 19.31 ± 0.86% under the same experimental conditions. Altogether, the above-mentioned results suggest that 5-(*N*-trifluoromethylcarboxy) aminouracil should exhibit radiosensitizing properties.

### 2.4. A Possible Mechanism of Electron-Induced Degradation of CF_3_CONHU

Without doubt, the C–F bond is one of the strongest single bonds involving carbon [30]. However, one can find examples of reactions of trifluoromethylated aromatic compounds resulting in an expulsion of fluoride anions. One of them concerns the hydrolysis of hydroxybenzotrifluorides and fluorinated uracil derivatives showing the impact of a deprotonation at the N1 and N3 positions on the labilization of one of the fluoride atoms within the CF_3_ group [31]. The other work shows various examples of reactions of trifluoromethylated compounds with different nucleophilic agents [32]. Not only hard nucleophiles can be responsible for the CF_3_ group decomposition. Trifluorothymidine treated with a phosphate buffer in an aqueous solution affords 5-carboxy-2’-deoxyuridine in a very good yield [33]. Moreover, trifluoroacetates undergo hydrolysis in mild basic conditions [34] or by sodium borohydride treatment [35].

In light of the aforementioned facts, one could expect the formation of either a deacetylation reaction product or hypothetical 2-oxazolidinone (see Appendix A) in the course of CF_3_CONHU radiolysis. Instead, *N*-uracil-5-yloxamic acid is generated (see Figure 5). The overnight conditioning of samples containing CF_3_CONHU in a basic aqueous solution, phosphate buffer or pure water confirmed that hydrolysis is not involved in the studied process and the formation of *N*-uracil-5-yloxamic acid occurs exclusively through the DEA process. To the best of our knowledge, there are no literature reports regarding the obtainment of oxalic acid derivatives starting from *N*-substituted trifluoroacetamides. Thus, one may treat the above-described protocol as a general approach that enables various oxalic acid analogues to be synthesized radiolytically.

In Figure 6 we present a computational proposal for the mechanism of the formation of *N*-oxamic acid derivative under the action of a solvated electron. In the first step, a solvated electron attaches to CF_3_CONHU and initiates the release of the fluoride anion from the primary radical anion (A → B, see Figure 6). The obtained difluoromethylene radical (B) reacts with water, giving, after hydrogen fluoride detachment, a fluorohydroxy intermediate (D). However, the aforementioned reaction with a water molecule turned out to be kinetically forbidden, as the activation barrier for water attachment was estimated to be as high as 37 kcal mol^−1^. Indeed, if the elemental reaction with such activation barrier is the bottleneck step, the reaction completion time (99% of substrate converted into product) amounts to ca. 30 million years in 298 K. To make sure that such a high kinetic barrier is not an artifact of the employed computational method, we recalculated it with the use of a popular B3LYP [36,37,38] and the MPWB1K [39] functional, dedicated to kinetic calculations (in both cases using the 6-31++G (d,p) basis set and the polarization continuum model (PCM) model of water), and with the G2MP2 [40] model of chemical accuracy, obtaining ΔG_B3LYP_* = 31.5, ΔG_MPWB1K_* = 39.3 and ΔG_G2MP2_* = 35.5 kcal mol^−1^. On the one hand this confirms the estimate obtained at the M06-2X/6-31++G (d,p) level and, on the other hand, indicates that the reaction of B with water is unlikely. Therefore, we proposed a hydroxyl anion rather than a water molecule as a nucleophile in the elemental reaction involving B (B → C, Figure 6), although the concentration of the former was only 10^−7^ M (pH = 7), while that of water amounted to 55.5 M under the experimental conditions. It turned out that B reacts with the hydroxyl anion without a kinetic barrier. More precisely, the optimization of the B···^−^OH complex, even for a distance between B and ^−^OH as large as 8 Å, converged spontaneously to intermediate C (Figure 6), and the thermodynamic stimulus associated with the formation of C was highly favorable (ΔG = −41.9 kcal mol^−1^). The subsequent elimination of fluoride anion from difluorohydroxy anion radical C proceeded with relatively low barriers (ΔG* = 10.7 and ΔG = 5.6 kcal mol^−1^), giving fluorohydroxy radical D. In the next step, a water molecule assisted in a concerted elimination of the last fluoride atom (in the hydrogen fluoride form) to produce radical E with an acceptable kinetic barrier (ΔG* = 24.6 kcal mol^−1^; D → E, Figure 6). Then, a water molecule attached to the radical center of intermediate E (with low activation barrier ΔG* = 10.3 and favorable thermodynamic stimulus ΔG = −19.2 kcal mol^−1^) and spontaneously transferred one of hydrogen atoms to the neighboring carbonyl oxygen, producing intermediate radical F (Figure 6). Such radical can be stabilized via the detachment of hydrogen atom by any encountered radical. For instance, if a *tert*-butanol radical (*tert*-butanol radicals (*tert*-But^•^) are formed in the amount equivalent to that of the ^•^OH radicals during the reaction between the hydroxyl radicals (from water radiolysis) and *tert*-butanol used as an ^•^OH radical scavenger [8]) occurs in the proximity of F, it spontaneously (barrier-free) detaches the hydrogen atom from F, giving closed shell products: a *tert*-butanol molecule and the experimentally observed *N*-uracil-5-yloxamic acid (F → G, Figure 6). The complex of a radical F and *tert*-But^•^ is so reactive that its geometry optimization, even for a distance of 10 Å (between radical centers of F and *tert*-But^•^ molecules), leads barrier-free to the product complex. Due to the above-mentioned fact, the thermodynamic stimulus for the final step, ΔG_sep_ = −67.2 kcal mol^−1^, was estimated for the separated reagents (F + *tert*-But^•^ → G + *tert*-butanol).

However, one could still argue that despite the fact that reaction B → C is barrier-less when a hydroxyl anion is a nucleophile, a much smaller concentration of ^−^OH compared to that of water makes the rate of this reaction insufficient for the whole process to be completed in a reasonable time. The numerical integration of the set of appropriate kinetic equations (see Figure 6) revealed that the formation of *N*-uracil-5-yloxamic acid was completed in ca. 7 h (see Appendix A and Figure 7) which, at least intuitively, seems to be reasonable. It is worth noticing that the DFT model does not possess chemical accuracy, and the activation barrier of a bottleneck step (D → F, Figure 6) lowered by only 1 kcal mol^−1^ led to ca. 0.6 h for the reaction completion time, which additionally shows that the proposed mechanism may indeed be operative. 

### 2.5. Cytotoxicity

A good radiosensitizer should be characterized by low cytotoxicity. In order to determine the cytotoxicity of CF_3_CONHU toward PC3 (prostate cancer cells) and HDFa (normal fibroblasts) lines, the WST-1 assay has been performed. This test is based on the conversion of a tetrazolium salt to a chromatic formazan in the presence of intracellular mitochondrial dehydrogenase. The amount of dye generated by the enzyme is directly proportional to the amount of living cells [41].

CF_3_CONHU has been tested at seven concentrations: 0 (for control), 10^−4^, 10^−5^, 10^−6^, 10^−7^, 10^−8^ and 10^−9^ M; and two incubation time variants: 24 and 48 h (Figure 8). The data show a statistically significant reduction of viability in two cases. When it comes to the PC3 line, it is observed at the 10^−4^ M concentration in a 48 h time variant (the reduction of viability to 90.79 ± 1.3%). The extension of the incubation time of PC3 cells with the tested uracil derivative to 72 and 96 h results in a similar effect—the obtained results (see Appendix A) show that the tested derivative causes a statistically significant reduction in cell viability compared to the control (up to 86.5 ± 1.6% for 72 h incubation and up to 87.3 ± 1.7% for 96 h incubation) only for the highest concentration tested (10^−4^ M). The curves of proliferation are presented in the Appendix A. In the case of the HDFa line, the statistically significant reduction of viability to 92.53 ± 1.4% is observed also at 10^−4^ M, in the 48 h time variant. The presented data show that the cytotoxicity of the studied compound is very low for both lines and, furthermore, no meaningful differences have been observed between both lines.

### 2.6. Clonogenic Assay

The radiosensitivity of PC3 cells treated with the studied compound was determined by clonogenic assays. The effect of CF_3_CONHU on PC3 cells’ survival, when not combined with irradiation, was not observed. Figure 9 shows the change in cell survival with doses of 0.5, 1, 2 and 4 Gy in the presence of the tested derivative, compared to the survival of the control. The comparison of curves clearly shows that the presence of CF_3_CONHU enhances the radiosensitivity of the cells. The pre-treatment with 100 µM of the studied analogue reduced the survival of the PC3 cells irradiated with 0.5 Gy from 86.2 ± 2.2% to 68.8 ± 7.1%. With a somewhat higher dose (1 Gy), the surviving fraction was reduced from 69.5 ± 4.1% to 56.6 ± 1.9% after incubation. Also at higher doses of ionizing radiation, pre-treatment caused a significant reduction in the proliferation of PC3 cells (from 43.4 ± 2.8% to 32.5 ± 4.6% and from 11.5 ± 3.2% to 4.2 ± 1.3%, for 2 Gy and 4 Gy, respectively). The dose enhancement factor (ID_50_(−treatment)/ID_50_(+treatment)) [42], where ID_50_ means the radiation dose causing 50% growth inhibition) representing the radiosensitizing effect was equal to 1.52. Furthermore, based on these survival curves, the parameters for cellular radiosensitivity, such as α (coefficient for linear killing) and β (coefficient for quadratic killing) values, were calculated by a fitting using the linear–quadratic model (SF = exp(−1*(αD + βD2), where SF is the survival fraction and D is the dose). Average α and β values were 0.29 ± 0.015 and 0.064 ± 0.007, respectively, for the non-treated PC3 cells and 0.591 ± 0.092 and 0.027 ± 0.014, respectively, for PC3 cells treated with 10^−4^ M. In general, the incubation of cancer cells with CF_3_CONHU causes an enhancement of radiation-induced cancer cell killing, which suggests that the studied compound can be considered as a promising radiosensitizer.

### 2.7. Histone H2A.X Phosphorylation

Double-strand breaks generation is one of the most desirable type of DNA damage in the context of radiosensitization. Phosphorylation of histone H2A.X is the biomarker of such a damage. The flow cytometric assay was performed for human prostate cancer cells treaded with CF_3_CONHU at a concentration of 10^−4^ M and/or irradiated with a dose of 0 or 2 Gy in order to check the relationship between the formation of DNA double-strand breaks and the radiosensitivity of pre-treated PC3 cells. Our studies show that treatment with CF_3_CONHU results in a significant increase in the population of γH2A.X positive cells after irradiation with the dose of 2 Gy (Table 1 and Appendix A). The exposure of treated cells results in an enhancement of the γH2A.X level from 16.32 ± 0.12% to 37.48 ± 1.72%. This result suggests that the radiosensitivity of CF_3_CONHU-treated PC3 cells is related to (at least in part) the formation of DSBs.

## 3. Materials and Methods

### 3.1. Materials

5-Aminouracil, TFAA, TFA, streptomycin, penicillin, WST-1 and phosphate buffer were purchased from Sigma-Aldrich (Saint Louis, MO, USA). HDFa, F-12K, DMEM and fetal bovine serum (FBS) were obtained from Gibco (Life Technologies Limited, Paisley, Scots, UK). Thin-layer chromatography was performed on silica gel plates, 60G, F_254_ (Sigma-Aldrich, Saint Louis, MO, USA). The NMR spectra were recorded on a Bruker AVANCE III, 500 MHz spectrometer (Bruker, Billerica, MA, USA). Chemical shifts are reported in ppm relative to the residual signals of DMSO-d_6_ (2.49 ppm for ^1^H, 39.5 ppm for ^13^C). The radiolysis was carried out in a Cellrad X-ray cabinet (Faxitron X-ray Corporation, Tucsconcity, AZ, USA). HPLC analyses were performed on a Dionex UltiMate 3000 system with a diode array detector (Dionex Corporation, Sunnyvale, CA, USA). The LC-MS analyses were performed using a mass spectrometer TripleTOF 5600+ (ABSciex, Darmstadt, Germany) with UHPLC system Nexera X2 (Shimadzu, Canby, OR, USA). The absorbance was measured with an EnSpire microplate reader (PerkinElmer, Waltham, MA, USA). The colony size was assessed by using an inverted fluorescence microscope (IX73, Olympus, Tokyo, Japan).

### 3.2. Synthesis of 5-(N-trifluoromethylcarboxy)aminouracil

To a stirred solution of 5-aminouracil (100 mg, 0.79 mmol) in TFA (1 mL), TFAA (0.122 mL, 0.86 mmol) was added. The mixture was stirred for 24 h. After that time it was evaporated to dryness, and the crude product was purified with HPLC. The analytes were separated on a Synergy Polar-RP (Phenomenex) reverse-phase column. The isocratic elution with 5% phase B was used (mobile phase A: 0.1% formic acid and B: 80% ACN). The resulting product was obtained as a white solid (127 mg) in a 72.3% yield.

^1^H NMR (Bruker AVANCE III, 500 MHz, DMSO), δ: 11.40 (brs, 1H, NH), 10.79 (brs, 1H, NH), 7.79 (s, 1H, CH_Ar_); ^13^C NMR (125 MHz, DMSO), δ: 161.0, [156.68, 156.39, 156.10, 155.81 (q)], 150.8, 138.6, [119.68, 117.39, 115.09, 112.80 (q)], 109.5. HRMS (TripleTOF 5600+, SCIEX), m/z: [M–H]^−^ calcd for C_6_H_4_F_3_N_3_O_3_ 222.0131, found 222.0081; UV spectrum (water), λ_max_: 270 nm.

CAS Registry number: 31385-11-2.

### 3.3. Crystallography 

*XRD Measurements and Refinements.* A good-quality single-crystal of CF_3_CONHU was selected for the X-ray diffraction experiments at T = 295 K (Appendix A). It was mounted with an epoxy glue at the tip of glass capillaries. Diffraction data were collected on an Oxford Diffraction Gemini R ULTRA Ruby CCD diffractometer with MoKα (λ = 0.71073 Å) radiation. The lattice parameters were obtained by least-square fit to the optimized setting angles of the reflections collected by means of CrysAlis CCD [43]. Data were reduced using CrysAlis RED software^41^ and applying multi-scan absorption corrections (empirical absorption correction using spherical harmonics, implemented in the SCALE3 ABSPACK scaling algorithm). The structural resolution procedure was carried out using the SHELX package [44]. The structure was solved with direct methods that carried out refinements by full-matrix least-squares on F2 using the SHELXL-2017/1 program [44]. H-Atom bound to aromatic C-atom was placed geometrically and refined using a riding model with C–H = 0.93 Å and U_iso_(H) = 1.2U_eq_(C). All H-atoms bound to N-atoms were placed geometrically and refined freely with U_iso_(H) = 1.2U_eq_(N). All interactions were calculated using the PLATON program [45]. The ORTEPII, [46] PLUTO-78 [47] and Mercury [48] programs were used to prepare the molecular graphics.

Full crystallographic details of the title compound have been deposited in the Cambridge Crystallographic Data Center (deposition no. CCDC 2016475), and they may be obtained from: ccdc.cam.ac.uk, e-mail: deposit@ccdc.cam.ac.uk, or The Director, CCDC, 12 Union Road, Cambridge, CB2 1EZ, UK.

### 3.4. Radiolysis 

An aqueous solution of the studied compound at a concentration of 10^−4^ M was prepared. The mixture contained *t*-BuOH (0.03 M) and phosphate buffer (10^−2^ M, pH = 7.0). In order to deoxygenate the solution, argon was purged for ca. 3 min. All samples were subsequently IR-exposed with a dose of 140 Gy. The studied samples were prepared in triplicate, while the non-irradiated ones were used as controls.

### 3.5. HPLC and LC-MS

*HPLC analysis*. All CF_3_CONHU samples were analyzed with reversed-phase HPLC. For the separation of analytes, a C18 column (Wakopak Handy ODS, 4.6 × 150 mm, 5 μm in particle size and 100 Å in pore size) and an isocratic elution with 0.1% HCOOH flow rate 1 mL min^−1^ were used. The HPLC analyses were performed on a Dionex UltiMate 3000 System with a Diode Array Detector set at 260 nm.

*LC-MS analysis*. In order to identify the irradiation products, a Nexera X2 ultra high performance liquid chromatography (UHPLC) system coupled with a tandem mass spectrometer Triple TOF 5600+ (SCIEX) with a double electrospray interface operating in a negative mode has been used. Chromatographic separation has been performed on a C18 column (Kinetex Phenomenex, 2.1 × 150 mm^2^, 2.6 μm in particle size and 100 Å in pore size) at 25 °C, with an isocratic elution with 0.2% formic acid flow (0.3 mL min^−1^). The effluent was directed to waste for 1 min after each injection during analysis. MS parameters were as follows: the nebulizer gas (N_2_) pressure 30 psi, the spray voltage −4.5 kV, the flow rate 11 L min^−1^ and the source temperature 300 °C.

### 3.6. Calculations

A mechanism of a radiodegradation of CF_3_CONHU was studied computationally at the density functional theory level, using the M06-2X functional [49] and a 6-31++G (d,p) basis set [50,51]. The aqueous reaction environment was mimicked with the use of a polarization continuum model (PCM) [52]. All the geometries were fully optimized without any geometrical constraints, and the analysis of harmonic frequencies demonstrated that all of them were geometrically stable (all force constants were positive) or first-order transition states (all but one force constant positive). The Gibbs free energies of particular elementary reactions (ΔGs) were estimated as ΔEs (electronic energy change) between the product and substrate complexes corrected for zero-point vibration terms, thermal contributions to energy, the pV term and the entropy term. These terms were calculated in the rigid rotor-harmonic-oscillator approximation for T = 298 K and p = 1 atm [53]. For the last step, where the substrate complex is highly unstable and spontaneously transforms into products, ΔG_sep_ was calculated as the difference of the sum of Gibbs free energies of products and the sum of Gibbs free energies of substrates (each reagent optimized separately). Activation barriers, denoted as ΔG*, were calculated as the difference between the Gibbs free energies of transition state (TS) and the corresponding substrate complex. Each TS geometry was confirmed with the intrinsic reaction coordinate (IRC) procedure [54].

The adiabatic electron affinity, AEA_G_, was defined as the difference in Gibbs free energies of the neutral and anion radical at their corresponding fully relaxed geometries.

All calculations have been carried out with the Gaussian 16 package [55].

### 3.7. Cytotoxicity

In order to establish the level of cytotoxicity of the tested derivative, a WST-1 assay was performed. Normal human dermal fibroblasts (HDFa) were grown in a DMEM medium (high concentration of glucose and sodium puryvate), with antibiotics (streptomycin and penicillin) at a concentration of 100 U mL^−1^ and 10% FBS. The human prostate cancer cell line (PC3) was grown in a F-12K medium with 10% FBS and antibiotics (streptomycin and penicillin) at the same concentration. HDFa and PC3 cells were seeded into a 96-well plate in a density of 4 × 10^3^ per well and incubated in 37 °C and 5% CO_2_ overnight. Subsequently, the medium was replaced with the fresh one and the cells were treated with 5-(*N*-trifluoromethylcarboxy)aminouracil at 7 tested concentrations: 0 (for the control), 10^−4^, 10^−5^, 10^−6^, 10^−7^, 10^−8^ and 10^−9^ M. Next, treated cells were incubated (in the same conditions) with the studied compound for 24 and 48 h. After this time, the aqueous solution of WST-1 salt was added in an amount of 10 µL to each well and again incubated for further 4 h. The absorbance was measured at 440 nm. The vitality of control was taken as 100%. Received results were analyzed with the GraphPad Prism software. The statistical evaluation of the treated samples and untreated control was calculated using the one-way analysis of variance (ANOVA) followed by Dunnett’s multiple comparison test. The data were obtained from three independent experiments and each treatment condition assayed in triplicate. The differences were considered significant at *p* < 0.05.

### 3.8. Clonogenic Assay

PC3 cells (human prostate cancer cells obtained from Cell Line Service, Eppelheim, Germany) were treated with CF_3_CONHU in a concentration of 10^−4^ M and plated on 60 mm dishes in a density of 10^6^ cells per dish. After 48 h incubation (RPMI medium supplemented with 10% fetal bovine serum and with antibiotics (streptomycin and penicillin) at a concentration of 100 U mL^−1^, 37 °C, 5% CO_2_) the cells were exposed to 0.5, 1, 2 and 4 Gy, with the use of a Cellrad X-ray cabinet (Faxitron X-ray Corporation, 1.27 Gy min^−1^, 130 kV, 5.0 mA). After 6 h (the assay was performed in the variant with delayed plating after treatment to allow for repair processes [56]), the cells were trypsinized and plated on 100 mm dishes in a density of 800 cells per dish. The cell culture and irradiations have been performed under normoxic conditions. After 13 days, the formed colonies were fixed with 6.0% (v/v) glutaraldehyde and 0.5% crystal violet. The stained colonies were counted manually, and the colony size was evaluated using an inverted fluorescence microscope (Olympus, IX73, Tokyo, Japan). The data were analyzed, and survival curves were plotted following the linear-quadratic equation using the GraphPad Prism Software (San Diego, CA, USA). Plating efficiencies and survival fractions are shown in Appendix A.

### 3.9. Histone H2A.X Phosphorylation

To determine the level of DSBs caused by IR, the analysis of histone H2A.X phosphorylation, which is the marker of such a damage, was performed. PC3 cells were seeded at a density of 0.2 × 10^6^ per plate and incubated for 24 h (37 °C, 5% CO_2_). Next, the cells were treated with CF_3_CONHU at a concentration of 10^−4^ M and again incubated under the same conditions for 48 h. Upon this time, the plates with cells were irradiated with 0 and 2 Gy doses (1.27 Gy min^−1^, 130.0 kV, 5.0 mA). The incubation was extended for another hour (this time was optimized in previous experiments). Subsequently, the PC3 cells were dissociated with 1× Accutase solution and fixed, permeabilized and, in the last step, stained with anti-phospho-Histone H2A.X, FITC conjugate (the fixation, permeabilization and staining were performed according to the manufacturer’s protocol, FlowCellect^TM^ Histone H2A.X Phosphorylation Assay Kit, Merck, Hayward, CA, USA). The prepared cells were cytometrically analyzed (Guava easyCyte^TM^ 12, Merck, Kenilworth, NJ, USA). Nontreated cultures were the controls.

## 4. Conclusions

In this paper, we demonstrate that stationary radiolysis leads to the efficient radiochemical transformation of CF_3_CONHU to *N*-uracil-5-yloxamic acid and that CF_3_CONHU sensitizes prostate cancer cells (PC3 line) to X-rays.

The title compound belongs to the group of modified nucleobases, which undergoes DEA induced by solvated electrons. Synthesized and HPLC-purified CF_3_CONHU was dissolved in water and irradiated with a dose of 140 Gy of ionizing radiation, and the product analysis was carried out by mass spectrometry. It turned out that the attachment of a solvated electron to CF_3_CONHU leads to the generation of *N*-uracil-5-yloxamic acid. The decay of CF_3_CONHU (26.67 ± 0.31%), induced by ionizing radiation, was compared to the decomposition of BrdU (19.31 ± 0.86%) under the same experimental conditions. 

The mechanism of degradation of the studied reaction has been proposed using quantum chemical calculations carried out at the M06-2X/6-31++G (d,p) level. We demonstrated that the formation of *N*-uracil-5-yloxamic acid from CF_3_CONHU is allowed due to a barrier-less attack of a hydroxyl anion on the intermediate [CF_2_CONHU]^•^ radical. The rate constants for elementary reactions calculated using the M06-2X/6-31++G (d,p) activation barriers lead to a reaction completion time of the order of hours, which remains in accordance with the experiment.

A very low in vitro cytotoxicity of CF_3_CONHU was evaluated by the WST-1 assay. Finally, a clonogenic assay showed that the enhancement factor representing the radiosensitizing effect is equal to 1.52, and the cytometric analysis of histone H2A.X phosphorylation showed that the radiosensitizing effect of CF_3_CONHU is associated, at least in part, with the formation of double-strand breaks. Thus, one can reach the conclusion that CF_3_CONHU can act as a DNA radiosensitizer.

## Figures and Tables

**Figure 1 ijms-21-06352-f001:**
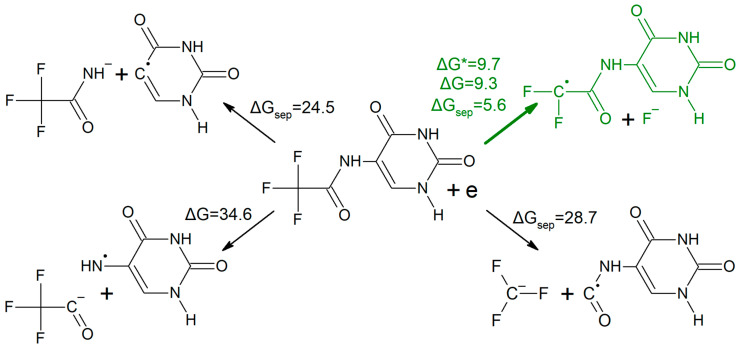
Energetics of dissociative electron attachment (DEA) processes in CF_3_CONHU, calculated at the M06-2X/6-31++G (d,p) level. The arrow associated with three values indicates the process that actually occurs in the studied system.

**Figure 2 ijms-21-06352-f002:**
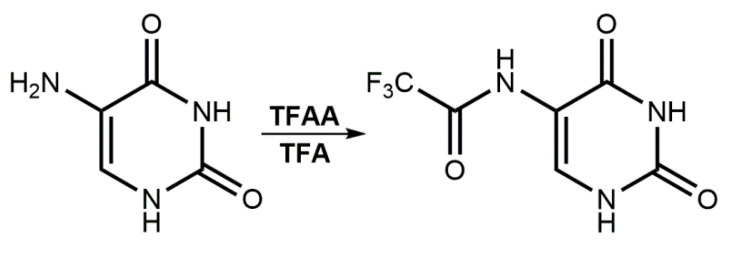
Synthesis of 5-(*N*-trifluoromethylcarboxy) aminouracil.

**Figure 3 ijms-21-06352-f003:**
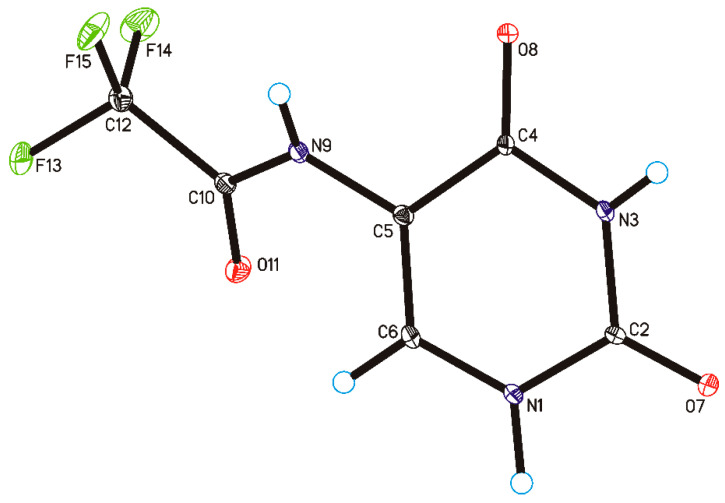
Molecular structure of 5-(*N*-trifluoromethylcarboxy) aminouracil, showing the atom-labeling scheme (displacement ellipsoids are drawn at the 25% probability level, and H-atoms are shown as small spheres of arbitrary radius).

**Figure 4 ijms-21-06352-f004:**
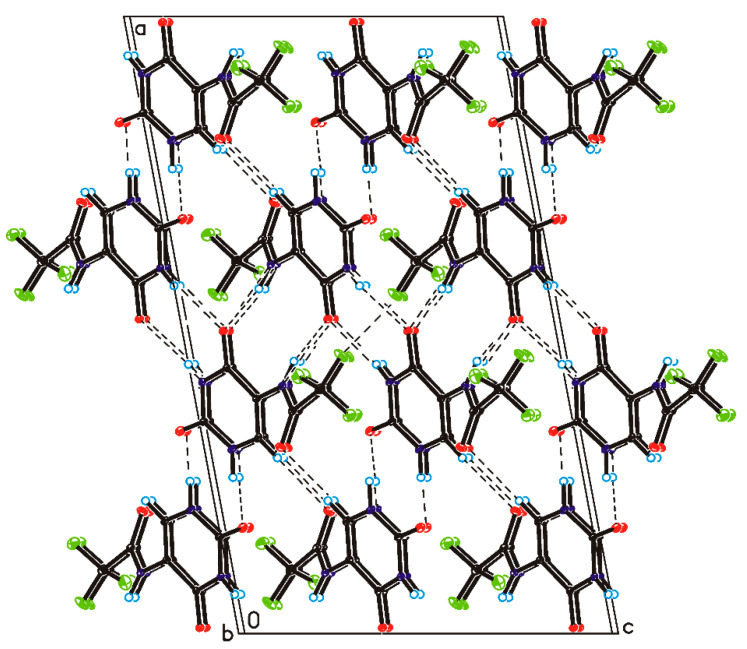
Crystal packing of 5-(*N*-trifluoromethylcarboxy) aminouracil viewed along the *b*-axis (hydrogen bonds are represented by dashed lines).

**Figure 5 ijms-21-06352-f005:**
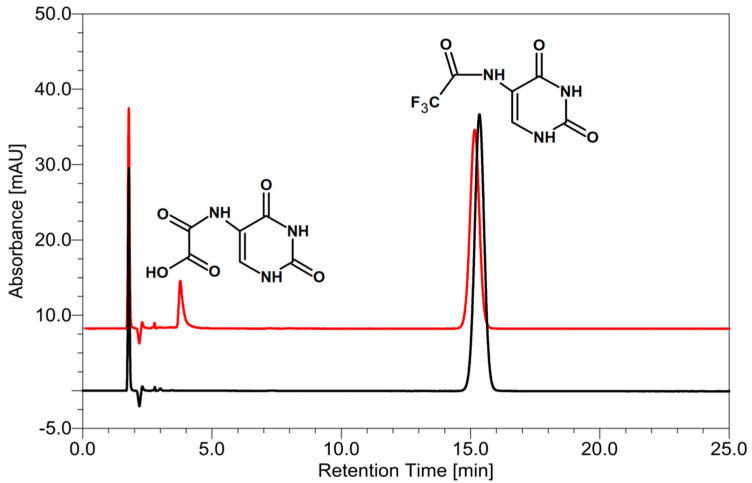
HPLC traces for 5-(*N*-trifluoromethylcarboxy) aminouracil radiolysis. Samples before (black) and after irradiation with a dose of 140 Gy (red). Chemical structures, as indicated by the LC-MS/MS analyses, are assigned to particular HPLC peaks.

**Figure 6 ijms-21-06352-f006:**
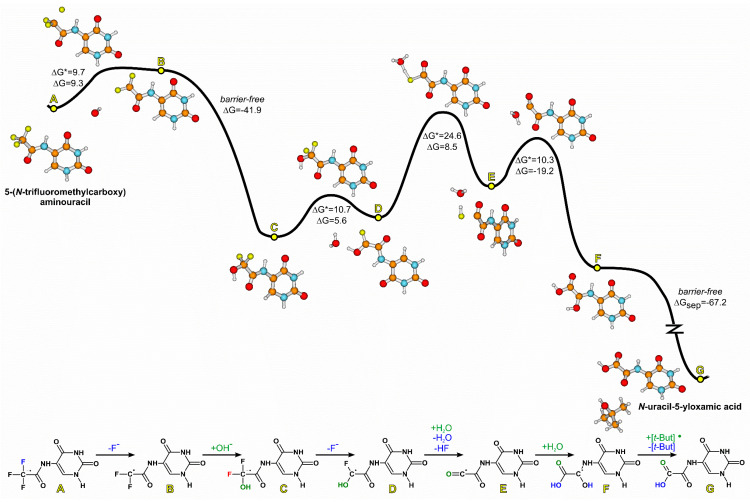
Gibbs free energy profile for the degradation of an anion radical of 5-(*N*-trifluoromethylcarboxy) aminouracil. Thermodynamic (ΔG) and kinetic (ΔG*) barriers in kcal mol^−1^.

**Figure 7 ijms-21-06352-f007:**
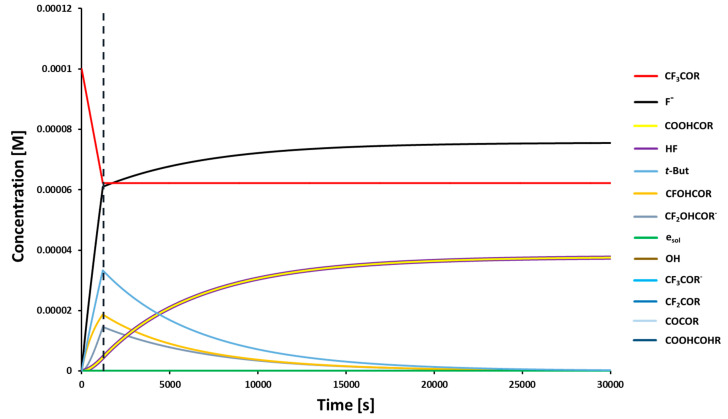
Kinetic curves for particular species involved in the radiolysis of a CF_3_CONHU solution (R = NHU). The vertical broken line indicates the moment at which the X-ray source was turned off. The system of kinetic equations (Appendix A) has been integrated for 3 × 10^4^ s.

**Figure 8 ijms-21-06352-f008:**
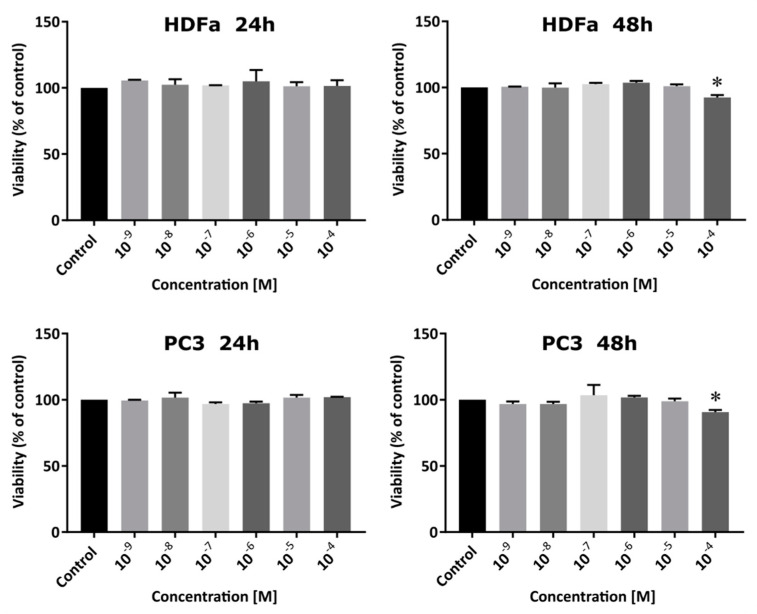
The viability of PC3 and HDFa cell lines after 24 and 48 h treatment with 5-(*N*-trifluoromethylcarboxy)aminouracil in a range of concentrations from 0 to 10^−4^ M. Results are shown as mean ± SD of three independent experiments performed in triplicate. * a statistically significant difference is present in the treated culture when compared with control (untreated culture).

**Figure 9 ijms-21-06352-f009:**
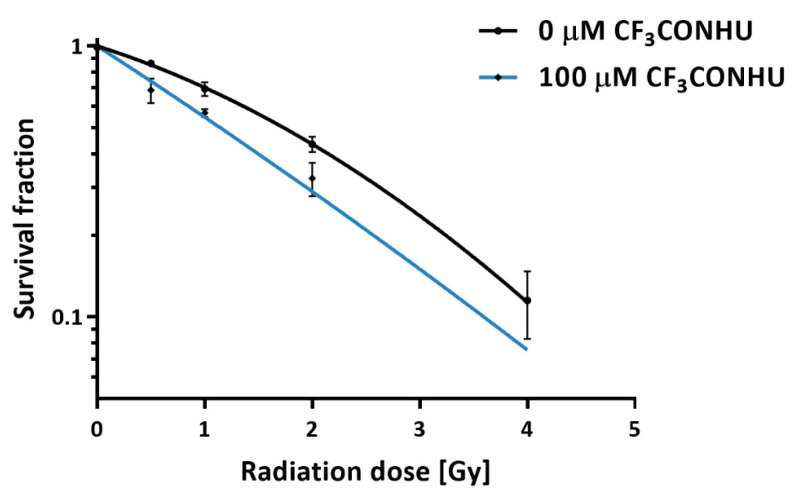
Effect of 100 µM 5-(*N*-trifluoromethylcarboxy) aminouracil treatment combined with irradiation on the survival of PC3 cells. Data obtained with the clonogenic assay. Experiments were performed at least three times, and the results are expressed as mean ± standard deviation. The average plating efficiencies for the controls with and without pretreatment are equal to 40.66% (0 µM), and 42.09% (100 µM)

**Table 1 ijms-21-06352-t001:** Fractions of γH2A.X positive cells 1 h after irradiation. Results are shown as mean ± standard deviation (SD) of two independent flow cytometry experiments.

Dose [Gy]	% of γH2AX Positive Cells	
	Control (Non-Treated)	CF_3_CONHU (10^−4^ M, 48 h)
**0 Gy**	6.91 ± 0.17	7.00 ± 0.55
**2 Gy**	16.32 ± 0.12	37.48 ± 1.72

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
