# Peer review of "5-(N-Trifluoromethylcarboxy)aminouracil as a Potential DNA Radiosensitizer and Its Radiochemical Conversion into N-Uracil-5-yloxamic Acid"

_ijms, 2020, doi:10.3390/ijms21176352_

Round 1

Reviewer 1 Report

This manuscript describes the synthesis and reactions of a new radiosensitizer structurally related to 5-Bromo-uracil with F3C-C(O)NH replacing Br at the 5-position of uracil. The energetics of forming a reactive nucleobase radical upon irradiation were calculated and the molecular and crystal structure s were determined. Finally, the effect of the substance on cell proliferation was tested in fibroblasts and the prostate cancer cell line, PC3, and the radiosensitizing effect on clonogenic cell inactivation was demonstrated in PC3 cells. The paper is well written and based on a sound rationale. The chemistry aspects seem to be expertly performed. However, the radiobiology aspects have a number of shortcomings that will need a major revision.

In the introduction, the radiobiological rationale is presented as the need for radiosensitisers to overcome radiation resistance due to hypoxia. Here, the oxygen fixation hypothesis (see Hall & Giaccia, Radiobiology for the Radiologist, 7th ed. 2012; M.C.  Joiner, chapter 4 in Joiner & van der Kogel (eds), Basic Clinical Radiobiology, 5th ed. 2019) should be included in the overview, and the classification of radiosensitiser should distinguish between hypoxic radiosensitiser and radiosensitiser that do not rely on hypoxia. In line with the proposed hypoxic rationale, the radiosensitiser was designed to react with solvated electrons (a product of radiolysis) which are present mainly under hypoxia whereas they are scavenged by molecular oxygen to form superoxide radicals under oxic conditions. While radiolysis of the radiosensitiser was performed under hypoxia (line 146), no information on oxygenation status for cell irradiations is given in the Materials and Methods. Based on the rather high radiosensitivity of the PC3 cells in Fig. 9, it seems possible that the cells were irradiated under oxic conditions. This needs to be clarified because the use of oxic conditions will impact on the rationale for the experiments and the interpretation of the results.

Radiolysis under hypoxia was performed with a dose of 140 Gy, resulting in a yield of the radiosensitiser of 26.7 pct. However, this dose is 50-100 fold larger than the doses in the cell survival experiments so the clinically relevant yield of the reactive radical would be expected to be of the order of 0.3-0.5 pct, and even smaller in the presence of oxygen.  The authors do not show that the free radical formed by the reaction with a solvated electron under hypoxia actually attacks DNA to produce strand breaks. Can they exclude that the radiosensitiser may be incorporated into DNA and produce complex strand breaks upon irradiation, similar to the mode of action for BrdU ? The authors should test if the actual mechanism is consistent with their rationale for the radiosensitiser since the latter mode would not depend on the level of oxygenation.

Section 2.5 on cytotoxicity uses the WST-1 assay which produces a colour change in cells with intact mitochondria, and thus the intensity is a surrogate for the number of cells. Therefore, the assay can be used to monitor cell proliferation or cell death provided that the investigated substance does not interfere with WST-1 or the mitochondria. The authors should show the changes in WST-1 signal as function of time, including the time point 0h in order to assess whether cells proliferate (cell number goes up), are inhibited (cell numbers stay constant), or suffer cytotoxic death (cell number goes down). In order to detect an effect on proliferation, 24-48h incubation is too short to detect small or moderate effects, usually 72-96h will be needed. Furthermore, in the case of a cytotoxic effect, this might well act via the mitochondria. Potential reactions of the radiosensitiser with the redox system or with WST-1 itself should also be considered. In relation to the mode of action, a design where the radiosensitiser is added for a fixed period of time, e.g. 24h, followed by incubation without the radiosensitiser would be easier to interpret. Why was the highest – slightly ‘toxic’ – concentration used for the cell survival experiments ?

The presentation of the survival data in two graphs in Fig. 9 is redundant and the linear y-axis is not well suited for analyzing radiosensitivity. Survival curves should be analysed using the linear-quadratic model, -ln(SF)=aD+bD2,  and presented in a single, semi-logarithmic plot of the surviving fraction (SF) as function of dose (see Hall & Giaccia, Radiobiology for the Radiologist, 7th ed. 2012; M.C.  Joiner, chapter 4 in Joiner & van der Kogel (eds), Basic Clinical Radiobiology, 5th ed. 2019). In doing so, it is important to calculate mean values on the basis of the ln(SF), i.e. to use geometric and not the arithmetic mean values. Standard deviations and standard errors of means will then be asymmetric in a linear plot but symmetric in the semi-logarithmic plot. In addition to this, the coefficients, a and b, of the linear-quadratic fits should be determined by fitting survival data from each individual experiment in order to calculate mean values and standard errors of means. This will allow an assessment if the radiosensitiser affects the linear or quadratic components of inactivation.

The plating efficiency (PE) for unirradiated cells should be given in the main text and the colony yields and SF included in the data table in the Supplementary Materials. Were plates seeded in triplicate and how many independent replicate experiments were performed ? The clonogenic assay was performed with 6h delayed plating after irradiated and a fixed number of cells per dish. With 800 cells seeded and PE~0.4, more than 300 colonies would be counted on the dish. Such high colony densities may lead to incorrect scoring. The 6h delay after irradiation implies that DSB repair will be incomplete since complex lesions may require up to 24h for complete repair. The potential influence on the results should be discussed. The Materials and Methods describes that colony size was evaluated under an inverted fluorescence microscope. How many cells defined a colony, and why was a fluorescence microscope used to score colonies stained with crystal violet ? Finally, the PC3 cell line is normally very radioresistant with doses for SF=0.1=10 pct (D10) in the range of 6-8 Gy. Why were the PC3 cells much more radiosensitive (D10~4 Gy) in the present study ?

Reviewer 2 Report

The manuscript entitled “5-(N-Trifluoromethylcarboxy)aminouracil as a potential DNA radiosensitizer and its radiochemical conversion into N-uracil-5-yloxamic acid” represents the results, which proof the efficiency of 5-(N-trifluoromethylcarboxy)aminouracil as radiosensitizer , working under hypoxia.

In my opinion, the synthesis, purification, crystallography, radiolysis and possible mechanism responsible for the formation of the observed radioproduct have been carried out by appropriate methods and are well supported by the results.

Additionally,  the cytotoxic test against the PC3 prostate cancer cell line and HDFa human dermal fibroblasts confirmed low cytotoxicity of the title 5-(N-trifluoromethylcarboxy)aminouracil.

The manuscript is built on a good body of research on this topic. It is also quite well written in clear and transparent language. The manuscript generally is prepared carefully. The title represents manuscript’s content. The authors clearly explain the intended practical application of the research.

There are few minor flaws:

  1. Row 7: In affiliation, Department of the Faculty should be added.
  2. Rows 55, 63, 67 and 166: The number of references should be given without spaces.
  3. Please add city and country of the manufacturer (Rows 284, 287-292 and 392) .
  4. As far as the title compound is a known substance, CAS Registry number of this compound should be added.
  5. Groups, from which the signals derive, should be added in 1H NMR spectrum.
  6. There are only two described signals for quaternary carbon atoms in 13C NMR spectrum. Please, mark all signals for quaternary carbon atoms in compound.

Besides above mentioned notes, I see no major flaws.

In my opinion, the research represented in the manuscript is interesting from scientific point of view and worth for publication after minor revision.
